# *Rhizopus oryzae* for Fumaric Acid Production: Optimising the Use of a Synthetic Lignocellulosic Hydrolysate

Reuben Marc Swart [ID], Hendrik Brink [ID] and Willie Nicol *[ID]

Department of Chemical Engineering, University of Pretoria, Lynnwood Road, Hatfield, Pretoria 0002, South Africa; reuben.swart@tuks.co.za (R.M.S.); deon.brink@up.ac.za (H.B.)
* Correspondence: willie.nicol@up.ac.za

**Abstract:** The hydrolysis of lignocellulosic biomass opens an array of bioconversion possibilities for producing fuels and chemicals. Microbial fermentation is particularly suited to the conversion of sugar-rich hydrolysates into biochemicals. *Rhizopus oryzae* ATCC 20344 was employed to produce fumaric acid from glucose, xylose, and a synthetic lignocellulosic hydrolysate (glucose–xylose mixture) in batch and continuous fermentations. A novel immobilised biomass reactor was used to investigate the co-fermentation of xylose and glucose. Ideal medium conditions and a substrate feed strategy were then employed to optimise the production of fumaric acid. The batch fermentation of the synthetic hydrolysate at optimal conditions (urea feed rate 0.625 mg L$^{-1}$ h$^{-1}$ and pH 4) produced a fumaric acid yield of 0.439 g g$^{-1}$. A specific substrate feed rate (0.164 g L$^{-1}$ h$^{-1}$) that negated ethanol production and selected for fumaric acid was determined. Using this feed rate in a continuous fermentation, a fumaric acid yield of 0.735 g g$^{-1}$ was achieved; this was a 67.4% improvement. A metabolic analysis helped to determine a continuous synthetic lignocellulosic hydrolysate feed rate that selected for fumaric acid production while achieving the co-fermentation of glucose and xylose, thus avoiding the undesirable carbon catabolite repression. This work demonstrates the viability of fumaric acid production from lignocellulosic hydrolysate; the process developments discovered will pave the way for an industrially viable process.

**Keywords:** xylose; lignocellulosic biomass; lignocellulosic hydrolysate; *Rhizopus oryzae*; fumaric acid; metabolic flux analyses

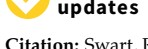



## 1. Introduction

There is a current global drive to move away from our reliance on fossil fuels and move toward renewable feedstocks for the production of chemicals. Fumaric acid is one of these chemicals being produced from petrochemical by-products, namely butane and benzene [1]. However, fumaric acid has been identified as a top value-added chemical that can be produced from sugars [2,3]. The current market size stands at USD 660.9 million per annum and is expected to grow at an annual rate of 5.5% for the period of 2021–2026 [4]. The industries that currently use fumaric acid include food and beverage, pharmaceutical, paper resins, unsaturated polyester resins, and animal feed [5–10].

*Rhizopus oryzae*, a zygomycete, has been found to be the most successful organism at producing fumaric acid when compared to a variety of natural and genetically modified organisms [5,11]. Our research group has published a number of articles on the production of fumaric acid with *R. oryzae* [12–17]. The optimum conditions for the production of fumaric acid have been identified. These conditions include the morphology, medium composition, growth procedure, pH, glucose feed rate, and urea feed rate. Industrially, however, it is unlikely that a pure glucose feed will be used for the production of fumaric acid as this would have to be sourced from cereal crops, thus encroaching on the food and animal feed industries. The favourable option would be to use a waste stream as feedstock. The lignocellulosic plant biomass is perfectly suited to the biorefining process for

the production of bio-based chemicals. Lignocellulosic material is comprised of cellulose, hemicellulose, and lignin, and it is often a waste stream for many processes, making it an ideal feedstock because it is inexpensive and renewable [18]. The hydrolysis of lignocellulosic biomass breaks down the polysaccharides into the monosaccharide units, of which glucose and xylose are the two predominant sugars [19,20].

Glucose has been a favourite of the fermentation industry since it is easily utilised by numerous microbes and is a widely available feedstock [20]. Glucose is consumed through the glycolytic pathway, which is common in many organisms, unlike the pentose phosphate pathway that is required for the fermentation of xylose [18]. This has led to the less frequent utilisation of xylose as a feedstock because the required pathways would have to be transferred to the identified organism [21]. However, in recent years, a drive towards more renewable feedstocks has grown. *R. oryzae* does have the ability to consume both xylose and glucose to produce fumaric acid [22]. This ability is highly beneficial as lignocellulosic hydrolysate can be used for the production of fumaric acid. The available literature is limited and only covers batch shake flask fermentations of xylose and glucose feedstocks [22–25]. The reported fumaric acid yields from these studies range from $0.31 \, \mathrm{g \, g^{-1}}$ to $0.58 \, \mathrm{g \, g^{-1}}$ (mass of fumaric acid produced per mass of substrate consumed), considerably lower than pure glucose fermentations that reach $0.93 \, \mathrm{g \, g^{-1}}$ [17].

The utilisation of xylose in the continuous production of fumaric acid as well as the co-fermentation of glucose and xylose need to be further understood. Our novel reactor and fermentation strategy can precisely control all critical medium conditions, allowing for the close monitoring of substrate consumption and metabolite production in order to uncover the physiology of *R. oryzae*.

This study aims to compare, analyse, and optimise the production rates and yields of fumaric acid achieved from the fermentation of pure glucose, pure xylose, and a synthetic lignocellulosic hydrolysate (LH). This will give greater insight into the utilisation of xylose as a substrate, the use of lignocellulosic hydrolysate (including only the predominant sugars) as a potential feedstock for fumaric acid production, and the effects that xylose has on the metabolism of *R. oryzae*.

## 2. Materials and Methods

### 2.1. Microorganism and Culture Conditions

*Rhizopus oryzae* (ATCC 20344), obtained from the Spanish collection of cultures (Colección Espanola de Cultivos Tipo, Valencia, Spain), was used for all the fermentations in this study. The culture was cultivated on potato dextrose agar and incubated at 30 °C for 5 days. The spores were suspended in sterile distilled water to achieve a spore concentration of $8 \times 10^6 \, \mathrm{mL^{-1}}$. A total of 10 mL of the spore solution was injected aseptically into each of the batch growth fermentations as the inoculum.

### 2.2. Medium

All the fermentations used the same mineral medium with the urea, and substrate concentrations varied depending on the experiment. The mineral medium contained the following (all values have units of $\mathrm{g \, L^{-1}}$): 0.6 $KH_2PO_4$, 0.507 $MgSO_4 \cdot 7 H_2O$, 0.0176 $ZnSO_4 \cdot 7 H_2O$, and 0.0005 $FeSO_4 \cdot 7 H_2O$ [26]. The biomass was grown under batch conditions with $3.1 \, \mathrm{g \, L^{-1}}$ glucose and $2.0 \, \mathrm{g \, L^{-1}}$ urea [12]. After the completion of the biomass growth phase with all glucose consumed, the reactor was drained and rinsed aseptically twice with the production medium to remove all nitrogen from the system. The immobilised biomass remained in the reactor. The batch production medium consisted of $20 \, \mathrm{g \, L^{-1}}$ glucose, xylose, or a glucose–xylose mixture as well as the mineral medium salts at the specified concentrations. The continuous production fermentations began with only the mineral solution. Urea was fed at a rate of $0.625 \, \mathrm{mg \, L^{-1} \, h^{-1}}$ for all production fermentations. The 50% mass-based glucose–xylose mixture (synthetic LH) was fed at a rate from $0.132 \, \mathrm{g \, L^{-1} \, h^{-1}}$ to $0.329 \, \mathrm{g \, L^{-1} \, h^{-1}}$ for the continuous fermentations. To achieve low dilution rates, high-concentration solutions of both the synthetic LH and urea were made with

$325.85\,\mathrm{g\,L^{-1}}$ and $16\,\mathrm{g\,L^{-1}}$, respectively. The dilution rate for the continuous production fermentations varied from $0.0018\,\mathrm{h^{-1}}$ to $0.0027\,\mathrm{h^{-1}}$ and took into account the substrate and urea additions as well as the NaOH dosing. The urea solution incorporated the mineral solution to ensure that the mineral composition in the reactor remained constant over the duration of the experimental run. All the solutions were sterilised at $121\,^\circ\mathrm{C}$ for $60\,\mathrm{min}$. All chemicals used were obtained from Merck (Modderfontein, South Africa).

## 2.3. Fermenter Design and Operation

The reactor consisted of a glass tube and a stainless steel housing with a liquid volume of $1.08\,\mathrm{L}$. A rough polypropylene tube was inserted into the centre of the reactor onto which *R. oryzae* attaches. A scalpel was used to score the outer part of the tube and thereby create the attachment surface. The tube had a length of $386.5\,\mathrm{mm}$, with an internal and outer diameter of $32\,\mathrm{mm}$ and $40\,\mathrm{mm}$, respectively. Silicone tubing connected to the housing allowed for the recycling of the liquid phase, the gas phases, and the aseptic addition or removal of the chemicals. The reactor was sterilised at $121\,^\circ\mathrm{C}$ for $60\,\mathrm{min}$ before all fermentations. A gas mixture of 8% $CO_2$ and 16% $O_2$ with the complement $N_2$ was sparged constantly at a rate of $108\,\mathrm{mL\,min^{-1}}$ for all fermentations. The exhaust gas composition was analysed online with the Tandem Gas Analyser 0588 from Magellan Biotech (Borehamwood, UK). The liquid was recycled past an Endress + Hauser CPS171 pH-probe (Gerlingen, Germany), which measured the temperature and pH of the medium. The temperature was maintained at $35\,^\circ\mathrm{C}$. The pH of the batch growth fermentations was controlled and kept at pH 5, and the production fermentations were maintained at pH 4. The pH was controlled with the addition of a $10\,\mathrm{M}$ NaOH solution. The liquid flow rates of the sugar solution, mineral solution, and NaOH solution were recorded online. Since the reactor volume was limited to $1.08\,\mathrm{L}$, the inlet flow rate was equal to the outlet flow rate, since liquid was constantly added under continuous operation. This resulted in constant dilution. Further information on the reactor operation is described by Swart [15].

## 2.4. Analytical Methods

Samples were taken from the fermentations at varied increments to achieve a satisfactory resolution for the changing concentration profiles. The sampling intervals were determined from previous fermentations [15,17], and this was also iteratively corrected if the concentration profiles changed faster than expected. The sampling frequency was considered satisfactory if it produced a smooth concentration profile. The samples were analysed using High-Performance Liquid Chromatography (HPLC). The system used to analyse for glucose, xylose, and ethanol was the Agilent 1260 Infinity HPLC (Agilent Technologies, Santa Clara, CA, USA) equipped with a refractive index detector operated at $55\,^\circ\mathrm{C}$ and a $300 \times 7.8\,\mathrm{mm}$ Aminex HPX-87C column (Bio-Rad Laboratories, Hercules, CA, USA) operated at $60\,^\circ\mathrm{C}$. The mobile phase was a $0.005\,\mathrm{M}$ solution of $H_2SO_4$ with a flow rate of $0.6\,\mathrm{mL\,min^{-1}}$. For the analysis of glycerol and the organic acids (namely fumaric acid, malic acid, succinic acid, and pyruvic acid), the mobile phase was altered to $0.02\,\mathrm{M}$, with all other specifications remaining constant. It was discovered that the peaks of xylose and malic acid overlapped and could not be separated sufficiently with this system. In order to solve this, the concentration of malic acid was determined separately with the Waters HPLC (Waters, Milford, MA, USA) equipped with a UV-Vis detector and a $300 \times 7.8\,\mathrm{mm}$ Aminex HPX-87H column (Bio-Rad Laboratories, USA) operated at $35\,^\circ\mathrm{C}$. The mobile phase was a $0.02\,\mathrm{M}$ solution of $H_2SO_4$ with a flow rate of $1.0\,\mathrm{mL\,min^{-1}}$. Using the determined concentration of malic acid, this was subtracted from the combined peak of xylose and malic acid to determine the corrected concentration of xylose.

The dry cell mass was determined at the end of all experimental runs. Biomass measurements were not possible between the growth and production phases. The same biomass growth procedure was used for all experimental runs. Growth runs were terminated in order to determine the dry cell mass after growth and before production. The biomass was removed from the polypropylene, washed with $1\,\mathrm{L}$ of distilled water, and then filtered

through a 110 mm Grade 541 Whatman filter paper. The filter paper and biomass were then dried at 70 °C for 48 h before being weighed.

*2.5. Production Rate and Yield Consolidation*

Because the reactor was operated with a constant feed of liquid, it had a dilution rate, and therefore the concentration profiles could not be directly used to determine the production rates. To determine the production rates from the concentration profiles, Equation (1) was used [27]. This differential equation calculates the molar change of a species in the reactor by accounting for the entry, exit, production, or consumption of a species. The desired variable is $r_j$; Equation (1) was therefore reworked into Equation (2). $\frac{dN_j}{dt}$ was calculated using the concentration profiles obtained from the HPLC analysis. Equation (3) illustrates how $\frac{dN_j}{dt}$ was calculated. It was assumed that the differential molar change term and the difference molar change terms were approximate for the calculations. The concentrations in between sample values were interpolated to calculate the difference. Equation (1) was solved using the Euler integration with a time increment of 1 s; this was the same increment that all other online measurements were sampled at. The effluent volumetric flow rate, $Q_e$, comprised of the volumetric feed rate and the volume sampled from the fermenter at specific times. All the production rates reported are average values over a minimum interval of 12 h.

$$\frac{dN_j}{dt} = Q_f C_j^f - Q_e C_j + r_j V, \tag{1}$$

$$r_j = \frac{\left( \frac{dN_j}{dt} - Q_f C_j^f + Q_e C_j \right)}{V}, \tag{2}$$

$$\frac{dN_j}{dt} \approx \frac{\Delta N_j}{\Delta t} = \frac{\Delta C_j}{\Delta t} V, \tag{3}$$

In order to confirm that all the metabolites had been accounted for, a mass balance was conducted for the systems. The total amount of substrate initially added or fed over the course of the fermentation was measured (this included glucose, xylose, and urea, depending on the specific fermentation conditions). The total molar amount of carbon added to the reactor was determined. The total amount of metabolites produced was then established by integrating the corrected production rates. The metabolites accounted for include fumaric acid, ethanol, malic acid, succinic acid, pyruvic acid, and glycerol. The total mass of the biomass produced during the production fermentation was identified and converted into carbon moles. The molar mass for *R. oryzae* and the process for determining the biomass produced is described by Swart [17].

Finally, the amount of carbon that exited the reactor as $CO_2$ had to be accounted for. Using the online gas $CO_2$ composition and the flow rate of the gas sparged into the reactor, together with Equation (4), the rate of $CO_2$ production was determined. The total molar amount of $CO_2$ produced was calculated by integrating the production rate with the fermentation time. The closure of the mass balance was determined by comparing the molar amount of carbon in the reactor with the sum of all the carbon accounted for at the end of the fermentation. The mass balance was conducted for all the fermentations, and it was confirmed that all carbon had been accounted for. This was claimed since all HPLC peaks were accounted for. A 10% tolerance was used to indicate a sufficient mass balance closure. The error was likely due to the assumption that the inlet and outlet gas flow rates were equal. Due to the pressure drop across the gas analyser, the outlet gas flow rate could not be measured.

$$r_{CO_2} = \frac{1}{V} \left( Q_{gas}(C_{CO_2}^o - C_{CO2}) - V_g \frac{dC_{CO2}}{dt} \right) \tag{4}$$

### 3. Results and Discussion

All the fermentations began with the same batch growth of biomass with excess nitrogen. The glucose concentration was used to achieve the correct thickness and covering of biomass on the polypropylene tube [12]. Once all the glucose was consumed, as indicated by online $CO_2$ production rates, the medium was drained, and the reactor was then rinsed and filled with the respective production medium in order to remove nitrogen from the reactor and induce the production of fumaric acid. The production fermentations were operated at pH 4, with a constant addition of urea at $0.625\,mg\,L^{-1}\,h^{-1}$. These variables were found to greatly affect the production of fumaric acid, with the values being the optimum operating conditions [17].

To investigate the feasibility of fumaric acid production with a lignocellulosic hydrolysate, it was first necessary to understand the metabolism of xylose and a glucose–xylose mixture. We conducted batch fermentations of glucose, xylose, and then a 50% glucose–xylose mixture that simulated lignocellulosic hydrolysate. The total sugar concentration for these fermentations was $20\,g\,L^{-1}$. Figure 1 shows the batch fermentation of glucose. It can be seen that the major products were fumaric acid and ethanol. The minor products of malic acid, succinic acid, and pyruvic acid reached maximum concentrations of $0.83\,g\,L^{-1}$, $0.37\,g\,L^{-1}$, and $0.137\,g\,L^{-1}$, respectively.

The production rate of fumaric acid and ethanol proved to be equivalent during the first 10 h of the fermentation, producing large amounts of ethanol—which is unfavourable. This production of ethanol was induced by the high glucose concentration. *R. oryzae* has been found to be a Crabtree-positive organism, producing ethanol under fully aerobic conditions [15]. Ethanol is an unwanted by-product because it decreases the yield of fumaric acid and complicates the downstream separation and processing. Although ethanol can be assimilated and metabolised, no fumaric acid is produced from it. Fumaric acid is produced by *R. oryzae* through a reductive tricarboxylic acid (TCA) cycle that is present in the cytosol [28]. Ethanol is likely consumed by its conversion to acetate—and afterwards to acetyl-CoA—from where it can be consumed by the TCA-cycle for the production of energy. Therefore, it offers no benefit to the production of fumaric acid.

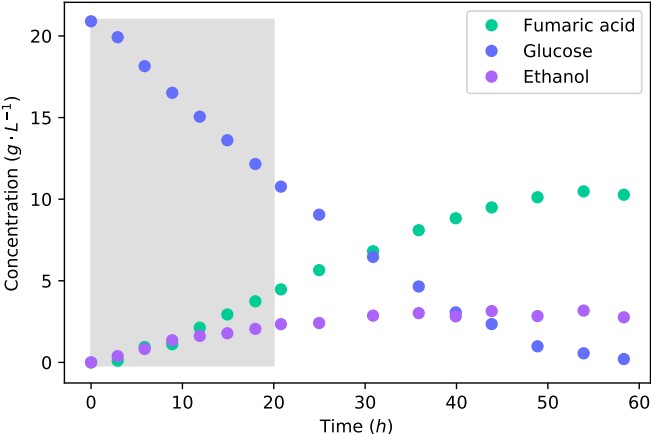

**Figure 1.** The concentration profiles for a batch fermentation of a $20\,g\,L^{-1}$ glucose solution. The shaded area indicates the 20 h interval used for metabolic flux calculations further on.

Using the equations and integration method outlined in Section 2.5, the effect of the dilution rate was accounted for to accurately determine the consumption and production rates of the organism. For the batch fermentation of glucose, the final accumulative yield of fumaric acid from glucose was $0.553\,g\,g^{-1}$. The yield of ethanol was found to be $0.191\,g\,g^{-1}$, which is a large yield for an unwanted by-product. It has been found that the production of ethanol could be avoided by carefully throttling the glucose feed rate on the reactor, which in turn increases the yield of fumaric acid from glucose up to $0.93\,g\,g^{-1}$ [15,17]. The high

yield of ethanol and low yield of fumaric acid in this batch fermentation clearly highlights the advantage of throttling the glucose feed rate in order to avoid the production of ethanol. The overall rate of fumaric acid production was $0.186 \, \text{g} \, \text{L}^{-1} \, \text{h}^{-1}$, and the maximum rate was found to be $0.291 \, \text{g} \, \text{L}^{-1} \, \text{h}^{-1}$. This maximum rate of fumaric acid production with the co-production of ethanol ($0.087 \, \text{g} \, \text{L}^{-1} \, \text{h}^{-1}$) is slightly less than the maximum fumaric acid production rate ($0.304 \, \text{g} \, \text{L}^{-1} \, \text{h}^{-1}$) found in a glucose-limited fermentation where no ethanol was produced [17]. The concentration of biomass and all other parameters were identical between these fermentations. The fumaric acid production can be considered to be equivalent for the two conditions. Because the batch fermentation has an unrestricted glucose intake, the rate of glycolysis increases to a point where the TCA-cycle reaches a limit. The residual carbon that cannot be accommodated through the TCA-cycle or fumaric acid production is directed to the production of ethanol. This illustrates the Crabtree effect.

Comparing the fermentation of glucose to that of xylose, as shown in Figure 2, it can firstly be seen that the duration of fermentation was considerably longer for xylose. Glucose fermentation ended after 58 h, whereas the fermentation of xylose took 166 h for the same mass of substrate. The average fumaric acid production rate was $0.073 \, \text{g} \, \text{L}^{-1} \, \text{h}^{-1}$, and the maximum rate achieved was $0.145 \, \text{g} \, \text{L}^{-1} \, \text{h}^{-1}$. Comparing the average production rates, the production rate of fumaric acid from xylose is 60.8% lower than that from glucose.

The metabolism of xylose is largely different to that of glucose. Xylose has to be catabolised to xylitol, then D-xylulose, and followed by D-xylulose-5-P, which enters the Pentose phosphate pathway from which D-glucose-6-P is produced; this is the start of glycolysis [29]. This is a longer pathway compared to the catabolism of glucose, which undergoes a single enzymatic step to produce D-glucose-6-P. These additional enzymatic steps required for the catabolism of xylose are likely the cause of the slow xylose utilisation and fumaric acid production. In a study on xylose utilisation by a recombinant *Saccharomyces cerevisiae*, it was found that the enzymatic route of xylose to glycolysis was the rate-limiting step which resulted in inefficient metabolism affecting the energy balance of the cell [30].

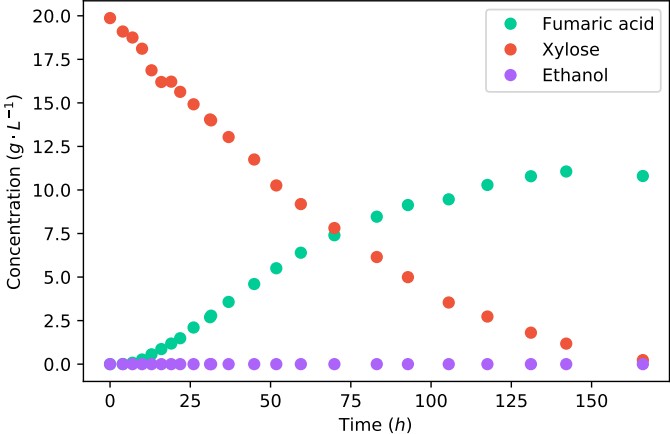

**Figure 2.** The concentration profiles of a batch fermentation of a $20 \, \text{g} \, \text{L}^{-1}$ xylose solution.

However, it can be seen in Figure 2 that there is no ethanol produced from xylose. This is a result of the lower glycolytic flux, which does not saturate the metabolism. All carbon can be accommodated by the reductive TCA cycle, producing fumarate, and the TCA cycle; therefore, no ethanol is produced as an overflow. The final accumulative yield of fumaric acid on xylose is $0.682 \, \text{g} \, \text{g}^{-1}$, which is considerably higher than that found for glucose (Figure 1). The increased yield was caused by the lack of ethanol production. This yield is also the highest yield of fumaric acid from xylose that has been reported in the literature—an improvement resulting from the operating conditions used. All the fermentations in the literature used $CaCO_3$ for pH control, which did not provide the optimal pH, and the carbon–nitrogen ratio was insufficiently controlled. These results indicate that xylose can

be a promising substrate for the production of fumaric acid. However, the co-fermentation of xylose and glucose is of key importance.

Figure 3 shows the concentration profiles of the fermentation of the synthetic lignocellulosic hydrolysate. Glucose is metabolised preferentially over xylose—in the first 21 h, glucose was consumed at a rate of $0.398 \, \text{g L}^{-1} \, \text{h}^{-1}$, while there was no consumption of xylose. This illustrates carbon catabolite repression (CCR), a well-known phenomenon that prioritises the most energy efficient substrate in a mixture and leads to a diauxic or two-phase utilisation of the substrate [18]. Once the glucose concentration was depleted, xylose consumption began and increased to a rate of $0.116 \, \text{g L}^{-1} \, \text{h}^{-1}$. In the pure substrate fermentations, the average rates of glucose and xylose consumption were $0.337 \, \text{g L}^{-1} \, \text{h}^{-1}$ and $0.107 \, \text{g L}^{-1} \, \text{h}^{-1}$, respectively. Thus, it can be seen that the catabolism of glucose was uninhibited by the presence of xylose; only after the complete consumption of glucose did xylose catabolism reach its full capability. The effect of the two-stage substrate utilisation could be plainly seen in the concentration profile of fumaric acid. While glucose was being consumed, the production rate was at $0.247 \, \text{g L}^{-1} \, \text{h}^{-1}$, which then dropped to $0.063 \, \text{g L}^{-1} \, \text{h}^{-1}$ once only xylose was remaining. It can, however, be seen that the ethanol produced during the catabolism of the glucose is now being consumed during the consumption of xylose.

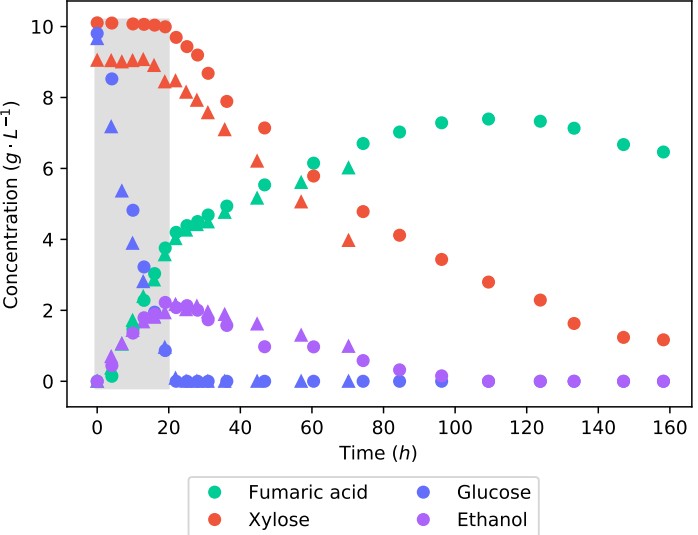

**Figure 3.** The concentration profiles of a $20 \, \text{g L}^{-1}$ 50% glucose and xylose batch fermentation. Two sets of concentration profiles can be seen: one shown with circles and the repeat shown with triangles. Good repeatability is demonstrated as the profiles follow near identical trends. The shaded area indicates a 20 h interval used for metabolic flux calculations further on.

The fumaric acid yield from the synthetic LH batch fermentation is $0.439 \, \text{g g}^{-1}$, which is considerably lower than the yields obtained from either of the pure substrate fermentations. This value is well within the range found in the literature [22–25]. The reason for this lower yield is likely a result of the co-fermentation of glucose and xylose. Different metabolic pathways are used to metabolise glucose and xylose; for this reason, the co-fermentation would require the production of more enzymes than necessary if only a single substrate was consumed. It can also be seen that ethanol is produced while glucose is being consumed, certainly contributing to the decreased fumaric acid yield. To make the production of fumaric acid from lignocellulosic hydrolysate viable, the yield will have to be improved. It has been found that minimising the medium glucose concentration negates the production of ethanol and drastically improves the yield of fumaric acid [15].

An effective method of controlling the glucose concentration is by beginning the production fermentation with a medium void of glucose. All glucose is then fed at a specific rate that is equal to the consumption rate. This allows for the rate of glycolysis to

be controlled, and we thus have the ability to negate the production of ethanol. Figure 4a–c shows the concentration profiles and substrate feed rate of the fermentation, where a 50% glucose–xylose mixture was continuously fed into the reactor. The fermentation began at a substrate feed rate ($0.132 \, \text{g} \, \text{L}^{-1} \, \text{h}^{-1}$) where an equivalent glucose feed rate did not produce any ethanol [15]. It can be seen that for 48 h, there was no production of ethanol; meanwhile, fumaric acid was still being produced. The substrate feed rate was then stepped up to $0.197 \, \text{g} \, \text{L}^{-1} \, \text{h}^{-1}$, immediately triggering the production of ethanol.

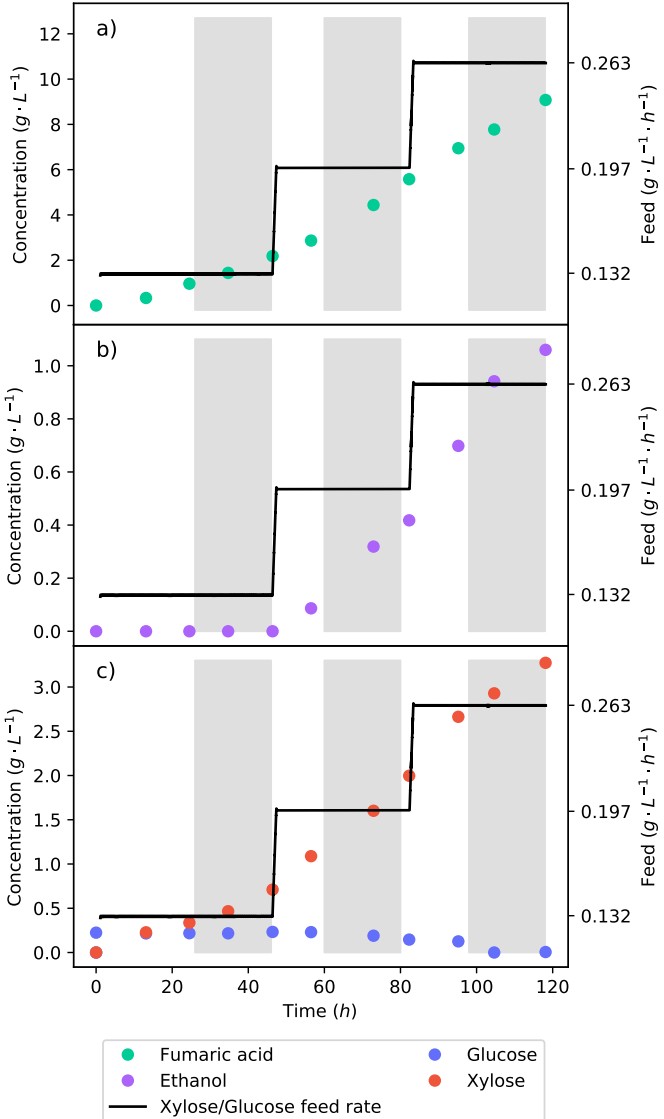

**Figure 4.** The continuous fermentation of a 50% glucose–xylose mixture. (**a**) The concentration profile of fumaric acid. (**b**) The concentration profile of ethanol. (**c**) The concentration profiles of glucose and xylose. The plot shows the feed strategy of the synthetic lignocellulosic hydrolysate in the reactor and the response of the metabolism to the change in substrate feed rate.

This indicates that the ethanol breakthrough rate is between $0.132 \, \text{g} \, \text{L}^{-1} \, \text{h}^{-1}$ and $0.197 \, \text{g} \, \text{L}^{-1} \, \text{h}^{-1}$. In pure glucose fermentations under the same conditions, it has been found that the ethanol breakthrough rate is between $0.263 \, \text{g} \, \text{L}^{-1} \, \text{h}^{-1}$ and $0.329 \, \text{g} \, \text{L}^{-1} \, \text{h}^{-1}$ [17]. The lessening of the ethanol breakthrough point is an unexpected effect, especially since it can be seen from the pure xylose fermentation (Figure 2) that no ethanol was produced. Ethanol is produced as a result of metabolite overflow (Crabtree effect), or for the production of energy. Adenosine triphosphate (ATP) levels have been found to decrease in the fermentation of

xylose as compared to that of glucose [30]. Therefore, the production of ethanol is likely a response to the lower ATP levels, causing glucose to be directed to ethanol in order to produce nicotinamide adenine dinucleotide (NADH) at a faster rate. Although the production of NADH from ethanol is less efficient, it is faster than the TCA cycle [27].

It can also be seen in Figure 4c that there is an accumulation of xylose from the lowest feed rate. There was, however, the complete consumption of glucose at all the feed rates tested. This results from CCR, where glucose is consumed preferentially over xylose. Using a 24 h running average at the end of the feed rate, it was found that xylose was consumed at a rate of $0.052 \, \mathrm{g \, L^{-1} \, h^{-1}}$, translating to 72.8% of the xylose fed being consumed. As the substrate feed rate was increased, the consumption rate of xylose also increased. Because the proportion of glucose to xylose in the feed remained constant, it can be seen that a higher glucose consumption rate enabled a higher xylose consumption rate. This was likely concurrent with the production of ethanol from the glucose, which provided more NADH. The production of ethanol is not a result of xylose accumulation since no ethanol was produced from the batch xylose fermentation.

The calculated yield of fumaric acid produced from the substrate—consumed after the first 48 h and at a feed rate of $0.132 \, \mathrm{g \, L^{-1} \, h^{-1}}$ where no ethanol was produced—was found to be only $0.425 \, \mathrm{g \, g^{-1}}$. The low yield is a result of a large portion of the substrate being directed to the TCA cycle for cell maintenance. The feed rates of $0.197 \, \mathrm{g \, L^{-1} \, h^{-1}}$ and $0.263 \, \mathrm{g \, L^{-1} \, h^{-1}}$ achieved yields of $0.693 \, \mathrm{g \, g^{-1}}$ and $0.483 \, \mathrm{g \, g^{-1}}$, respectively. This shows an initial increased yield with an increase in the feed rate. However, there is the production of ethanol. It has been found that the fumaric acid yield increases with an increased feed rate up to the point of ethanol breakthrough [17], after which the yield decreases. A feed rate of $0.164 \, \mathrm{g \, L^{-1} \, h^{-1}}$ was selected as a half-way point between $0.132 \, \mathrm{g \, L^{-1} \, h^{-1}}$ and the upper point of ethanol breakthrough ($0.197 \, \mathrm{g \, L^{-1} \, h^{-1}}$). The feed rate was tested for 48 h and is shown in Figure 5 by the triangular markers. Figure 5b shows that for the entire fermentation, no ethanol was produced; this indicates that the ethanol breakthrough point lies between $0.164 \, \mathrm{g \, L^{-1} \, h^{-1}}$ and $0.197 \, \mathrm{g \, L^{-1} \, h^{-1}}$. By negating ethanol production, the fumaric acid yield obtained at the end of the fermentation increased to $0.72 \, \mathrm{g \, g^{-1}}$.

Utilising the information gathered from the continuous fermentations where the feed rate was stepped, a strategy was hypothesised to increase the fumaric acid yield on a lignocellulosic hydrolysate feed. The production of ethanol can be avoided by controlling the feed rate on the reactor; all the glucose will be consumed, and the xylose will be allowed to accumulate. Once all the substrate has been fed, the substrate feed will be stopped, and the accumulated xylose will then be allowed to be metabolised. The same mass of substrate feed in the batch fermentations ($20 \, \mathrm{g \, L^{-1}}$) will be fed over the course of the fermentation. Figure 5 shows this fermentation.

For the first 24 h, the feed rate was at $0.132 \, \mathrm{g \, L^{-1} \, h^{-1}}$, allowing for the organism to adapt and for an inter-run comparison to be conducted. The feed rate was then increased to $0.164 \, \mathrm{g \, L^{-1} \, h^{-1}}$ for the remainder of the run until all the substrate had been fed. In Figure 5, it can be seen that the feed strategy was successful: no ethanol was produced, and once the feed rate stopped, the accumulated xylose was consumed. The fermentation was terminated once the production of fumaric acid ceased. The overall fumaric acid yield on the synthetic lignocellulosic hydrolysate feed was $0.735 \, \mathrm{g \, g^{-1}}$. Considering that the batch fermentation shown in Figure 3 has the same mass of substrate feed but a fumaric acid yield of $0.439 \, \mathrm{g \, g^{-1}}$, the benefit of controlling the metabolism is clear. Manipulating the substrate feed rate achieved a 67.4% improvement of the fumaric acid yield. The increased yield is a result of the negated ethanol production and the optimal metabolic flux that selects for the production of fumaric acid.

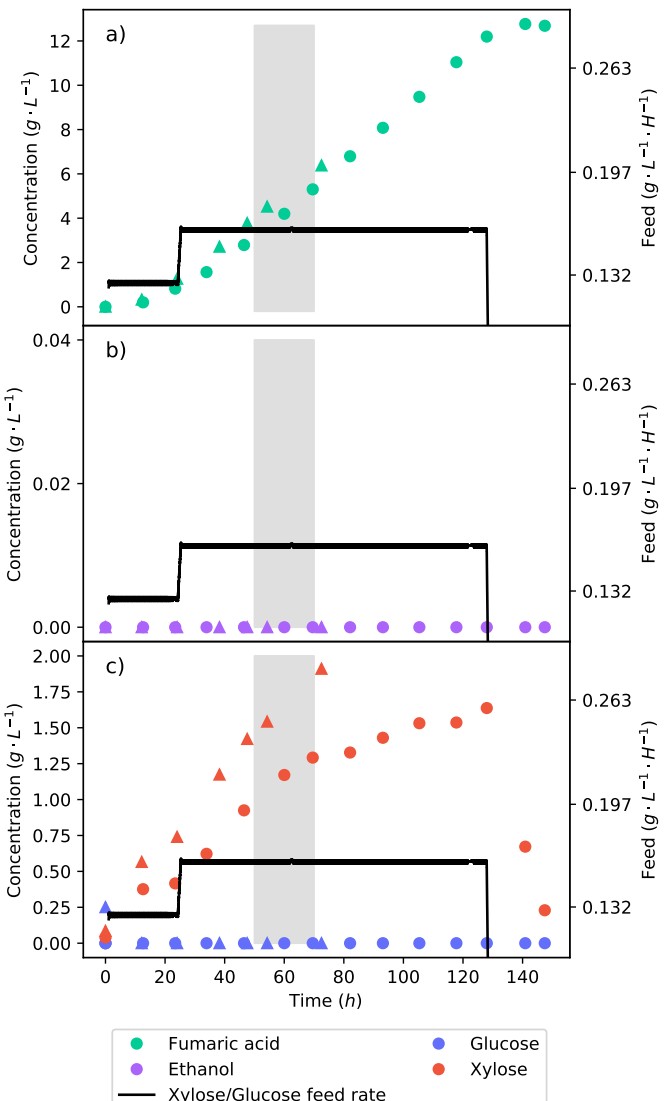

**Figure 5.** The continuous fermentation of a 50% glucose–xylose mixture fed at a rate of $0.164\,\mathrm{g\,L^{-1}\,h^{-1}}$. Two sets of concentration profiles can be seen: one shown with triangles (a preliminary fermentation) and the other shown with circles, which received $20\,\mathrm{g\,L^{-1}}$ of substrate over the fermentation. Good repeatability is demonstrated since the profiles follow nearly identical trends. The shaded area indicates a 20 h interval used for metabolic flux calculations further on. (**a**) The concentration profile of fumaric acid. (**b**) The concentration profile of ethanol. (**c**) The concentration profiles of glucose and xylose.

It was then considered whether a higher feed rate would produce a higher yield, as was later found by increasing the feed rate from $0.132\,\mathrm{g\,L^{-1}\,h^{-1}}$ to $0.164\,\mathrm{g\,L^{-1}\,h^{-1}}$. This increased the selectivity of carbon directed to fumaric acid. It was found that this relationship holds up to a glucose feed rate of $0.329\,\mathrm{g\,L^{-1}\,h^{-1}}$, which vastly improves the fumaric acid yield [17]. At this glucose feed rate, a fumaric acid yield of $0.93\,\mathrm{g\,g^{-1}}$ was achieved.

A fermentation of synthetic LH with this feed rate was then conducted, as shown in Figure 6. The feed rate was stepped up to $0.329\,\mathrm{g\,L^{-1}\,h^{-1}}$ after the first 24 h. The same mass of substrate ($20\,\mathrm{g\,L^{-1}}$) was to be fed; since the feed rate was far higher, this implied that the substrate would be delivered over a shorter period of time. Figure 6c shows that there was a considerable accumulation of xylose as a result of the high feed rate, which also had a clear effect on the production of ethanol (Figure 6b). In contrast, the glucose concentration remained low, indicating that the feed rate was matched by the rate of consumption. It can



be seen that the production of fumaric acid slowed down and then ceased 24 h later, after the feed rate halted (Figure 6a). This can also be seen in the lower feed rate fermentation in Figure 5, suggesting that the organism adapted to the co-fermentation of glucose and xylose in order to produce fumaric acid. Once glucose was no longer present, the production of fumaric acid stopped. The yield could be further improved if one were able to avoid xylose accumulation. However, this would require the ratio of glucose and xylose to be tailored to the respective uptake rates, and this may not be possible with a hydrolysate. Table 1 summarises the crucial results from the fermentations with equivalent amounts of substrate. It can plainly be seen that the fermentation with the lower LH feed rate that avoided ethanol production outperformed the other strategies.

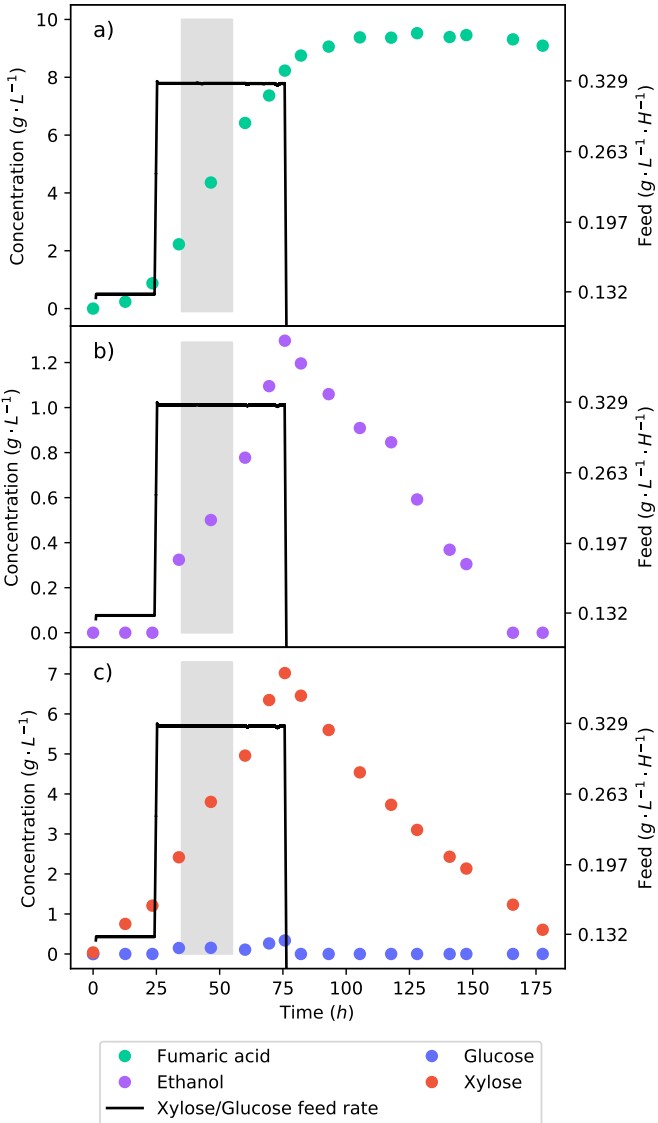

**Figure 6.** The continuous fermentation of a 50% glucose–xylose mixture at a feed rate of $0.329 \, \mathrm{g \, L^{-1} \, h^{-1}}$. The shaded area indicates a 20 h interval used for metabolic flux calculations further on. (**a**) The concentration profile of fumaric acid. (**b**) The concentration profile of ethanol. (**c**) The concentration profiles of glucose and xylose.

**Table 1.** Determining the effect of substrate and fermentation strategy on the yields, rates, and fermentation time. The following subscripts were used: S–substrate, F–fumaric acid, E–ethanol, G–glucose, and X–xylose.

| Run | $Y_{SF}$ [†] | $Y_{SE}$ [†] | $r_{F,max}$ * | $r_{F,avg}$ * | $r_{G,avg}$ * | $r_{X,avg}$ * | Run Time (h) | Mass Balance Error (%) |
|---|---|---|---|---|---|---|---|---|
| Glucose batch | 0.553 | 0.191 | 0.291 | 0.186 | 0.337 | - | 58.47 | 4.80 |
| Xylose batch | 0.682 | 0 | 0.145 | 0.073 | - | 0.107 | 166.05 | 5.60 |
| LH batch | 0.439 | 0.133 | 0.253 | 0.047 | 0.451 | 0.048 | 159.04 | 3.62 |
| LH High feed rate | 0.583 | 0.07 | 0.178 | 0.061 | 0.129 | 0.048 | 177.66 | 9.85 |
| LH Low feed rate | 0.735 | 0 | 0.146 | 0.096 | 0.076 | 0.066 | 148.40 | 9.25 |

[†] Accumulative yield over the run ($\mathrm{g\,g^{-1}}$). * Maximum rate calculated over a 12 h interval or the average rate over the entire run ($\mathrm{g\,L^{-1}\,h^{-1}}$).

Considering the repeatability of the fermentations presented, as visible in Figure 3, a duplicate of the fermentation was conducted. When comparing these two data sets, it can be seen that they are identical with all species following the same concentration profiles. Although the duplicate fermentation did not run to completion, it can still be said that the result is repeatable. The repeatability of the continuous fermentations has been proven in previous studies [15,17]; however, it will be discussed here for consistency.

All continuous fermentations were operated with the same conditions and substrate feed rate ($0.132\,\mathrm{g\,L^{-1}\,h^{-1}}$) for the first 24 h. Comparing the fumaric acid concentrations at the end of the 24 h, the mean was found to be $0.980\,\mathrm{g\,L^{-1}}$, with a standard deviation of 0.200; this resulted in a coefficient of variance of 0.204, which proves repeatability. For the 24 h duration of each of these runs, ethanol was expectedly not produced because the feed rate of $0.132\,\mathrm{g\,L^{-1}\,h^{-1}}$ was below the ethanol breakthrough point. This illustrates that the organism was operating in the same metabolic state for all four fermentations. Using the procedure outlined in Section 2.5, a mass balance was conducted over each of the fermentations in order to be certain that all the metabolites were accounted for. The mass balance compared the carbon added to the system to the sum of all the metabolites produced. It was found that the mass balance error for all the fermentations was less than 10%, indicating that the majority of the metabolites were accounted for. Table 1 reports the errors for the specific runs. The errors found are possibly a result of the outlet $CO_2$ flow rate that had to be assumed and could not be directly measured.

To gain further insight into the metabolism of *R. oryzae*, a metabolic flux model was developed for the metabolism of glucose and xylose. The metabolic flux model was verified by comparing the predicted $CO_2$ rates to that obtained from a mass balance. It was found that the metabolic flux model predicted the $CO_2$ production rates accurately, using the other known metabolite rates as input. Figure 7 shows the metabolic pathways determined for *R. oryzae*, metabolising glucose and xylose for the predominant production of fumaric acid, ethanol, and $CO_2$. The flux model was then solved for specific intervals, shown on the previous figures as shaded areas.

The flux model was solved with carbon balances as well as with NADH and NADPH balances. Further information on the development of the metabolic flux model and specific constants determined for *R. oryzae* are described by Swart [17]. Figure 8 shows the metabolic rates determined from the flux model for the batch fermentation of glucose (Figure 1), synthetic LH (Figure 3), and the optimal glucose continuous feed fermentation [17]. The result of a high glucose concentration is clearly demonstrated.

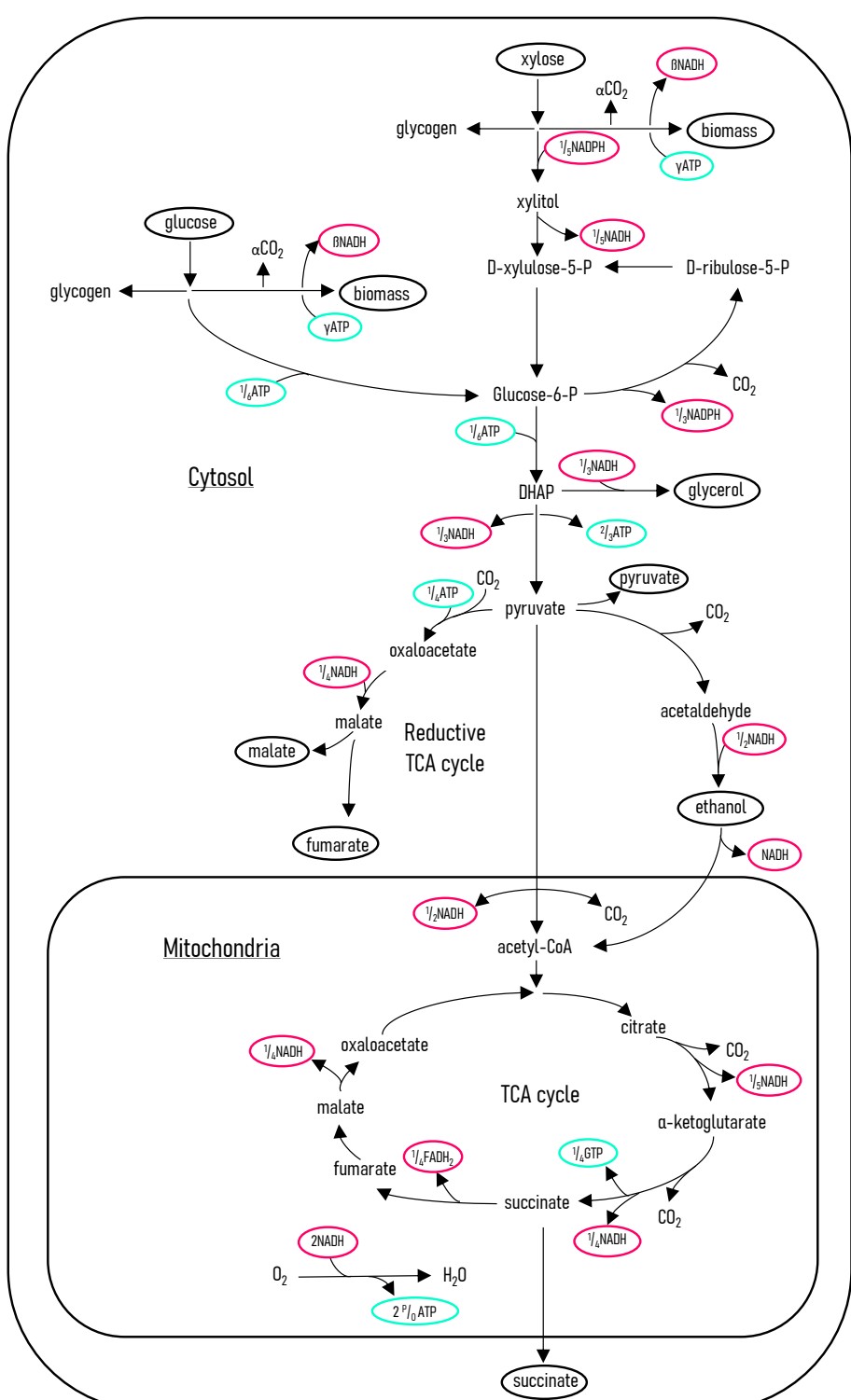

**Figure 7.** *R. oryzae* metabolic flux model. The metabolic pathways were determined by correlating a number of enzymatic studies of *R. oryzae* [28,29,31–33]. The flux model is written on the basis of carbon moles; this results in the illustrated fractional amounts of the energy-related compounds. The compounds that are either substrates or metabolites are circled in black.

In Figure 8a, it can be seen that the glucose uptake rates of both the pure glucose batch fermentation and the synthetic LH fermentation are equivalent. As a result of CRC, only glucose is consumed in the synthetic LH fermentation, indicating that xylose has no effect on the metabolism. It was found that the optimal glucose feed rate was below this maximum glucose uptake rate. Figure 8b shows the glycolytic flux of carbon to pyruvate;

this is the metabolic pathway after which the flux is split between the TCA cycle, fumaric acid production, and ethanol production. It can be seen that the glycolytic flux for both of the batch fermentations is higher than that of the continuous fermentation. Now, by comparing the ethanol production rates, it can be seen that the optimal glucose feed rate produced considerably less ethanol. This suggests that the production of ethanol is a result of a glycolytic threshold being surpassed. Once the glycolytic threshold has been passed, the proportion of carbon directed to ethanol increases, while fumaric acid production decreases. Operating below this glycolytic threshold improves both the yield and the rate of fumaric acid production.

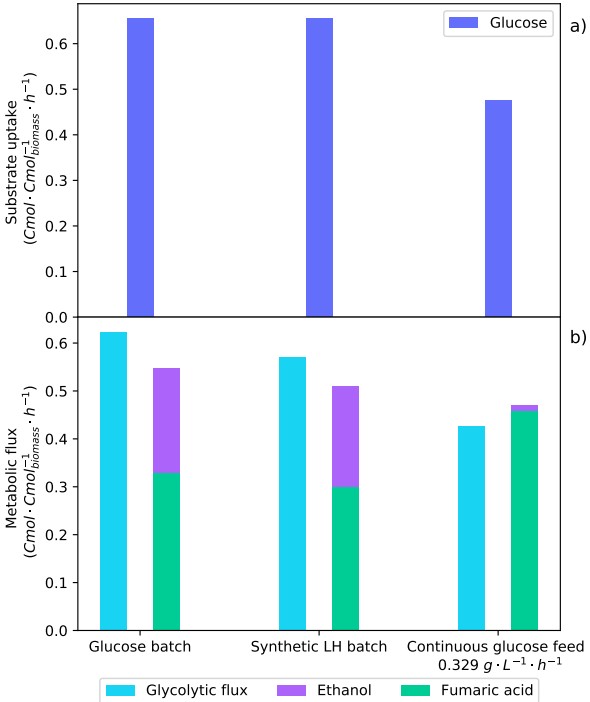

**Figure 8.** Metabolic flux rates determined for the batch glucose fermentation (Figure 1), the synthetic LH batch fermentation (Figure 3), and the optimal continuous glucose-fed fermentation [17]. The averaged metabolite rates from the shaded regions of these specific fermentations were used to solve the metabolic flux analysis. (**a**) The metabolic uptake rates of glucose and xylose (no xylose was consumed during these intervals). (**b**) The metabolic flux of carbon through the glycolytic pathway and the production rates of fumaric acid and ethanol.

The metabolic flux model was solved for each of the feed rates tested for the continuous synthetic LH fermentations. Figure 9 shows the metabolic fluxes determined. The co-fermentation of glucose and xylose can be seen for each of the feed rates in Figure 9a. A comparison of the glucose and xylose uptake rates shows a visible proportionality between the rates. An $R^2$ value of 0.983 was found for the first four substrate feed rates between $0.132 \, \text{g} \, \text{L}^{-1} \, \text{h}^{-1}$ and $0.263 \, \text{g} \, \text{L}^{-1} \, \text{h}^{-1}$. This is contrary to what was seen in the synthetic LH batch fermentation, where CRC resulted in the preferential consumption of glucose over xylose. The correlation of the glucose and xylose rates—considering that there was xylose accumulation at each feed rate—suggests that there is a dependency of xylose on glucose. An increased glucose feed rate enables a higher xylose uptake rate. Considering the feed rate of $0.329 \, \text{g} \, \text{L}^{-1} \, \text{h}^{-1}$, it can be seen that the proportionality between the glucose and the xylose rates no longer holds. The glucose consumption rate has increased proportionally with the increased feed rate; however, the rate of xylose uptake decreased. Considering the glycolytic flux in Figure 9b for the substrate feed rates of $0.263 \, \text{g} \, \text{L}^{-1} \, \text{h}^{-1}$ and $0.329 \, \text{g} \, \text{L}^{-1} \, \text{h}^{-1}$, it can be seen that they are equivalent rates. This suggests an upper limit for the glycolytic flux during the co-fermentation of glucose and

xylose. Once the upper limit is reached, glucose is used preferentially over xylose, which results in a decreased xylose consumption rate.

In Figure 9b, a feed rate of $0.164\,\mathrm{g\,L^{-1}\,h^{-1}}$ is shown to be the optimum, directing the highest fraction of carbon consumed to fumaric acid. The feed rate below ($0.132\,\mathrm{g\,L^{-1}\,h^{-1}}$) has a high fraction that is directed to the TCA cycle, which results in a low fumaric acid yield. This low yield is overcome when the feed rate is increased, and the fraction of carbon directed to the TCA cycle accordingly decreases. The higher feed rate ($0.197\,\mathrm{g\,L^{-1}\,h^{-1}}$) surpasses an upper threshold of the glycolytic flux and induces the production of ethanol, decreasing both the yield and the rate of fumaric acid production. Comparing the glycolytic flux at which the ethanol breakthrough occurs for the pure glucose fermentation (Figure 8b) and that of the synthetic LH fermentations (Figure 9b), it can be seen that ethanol production starts at a lower glycolytic flux during the co-fermentation of glucose and xylose. This suggests some effect that xylose has on the glycolytic flux and supports the evidence indicating that xylose causes an inefficient metabolic state, which in turn affects the energy balance [30]. This energy imbalance would explain why the ethanol breakthrough occurs at a lower substrate uptake rate.

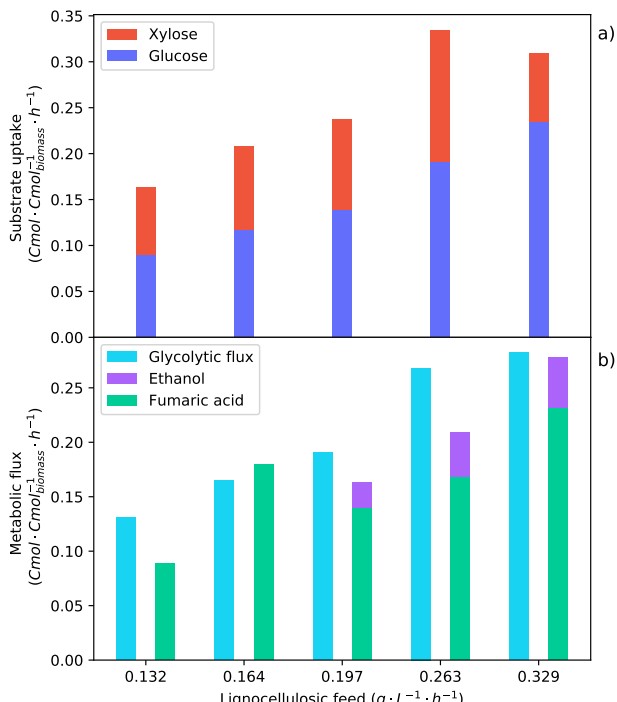

**Figure 9.** Metabolic flux rates determined for the specific synthetic LH feed rates that were indicated by the shaded intervals in Figures 4–6. The averaged metabolite rates from the shaded regions of the specific feed rates were used to solve the metabolic flux analysis. (**a**) The metabolic uptake rate of glucose and xylose. (**b**) The metabolic flux of carbon through the glycolytic pathway and the production rates of fumaric acid and ethanol.

## 4. Conclusions

The production of fumaric acid from glucose is a well-studied topic, whereas the more industrially viable option to use lignocellulosic hydrolysate has had little attention. Utilising a novel bioreactor and optimal medium conditions, we studied the use of xylose and a synthetic lignocellulosic hydrolysate for the production of fumaric acid. The highest known yield of fumaric acid on xylose was achieved ($0.682\,\mathrm{g\,g^{-1}}$) in a batch fermentation, which we attribute to the closely controlled and optimal medium conditions. In a batch fermentation of the synthetic lignocellulosic hydrolysate, it was found that the high concentration of glucose induced an overflow mechanism, causing ethanol production which greatly affected the yield ($0.439\,\mathrm{g\,g^{-1}}$). Utilising continuous fermentation with a low feed rate

$(0.164\,\mathrm{g\,L^{-1}\,h^{-1}})$ for the glucose–xylose mixture, the metabolism was controlled at an optimum point in order to select for the production of fumaric acid and simultaneously negate ethanol production. This greatly improved the fumaric acid yield on the substrate to $0.735\,\mathrm{g\,g^{-1}}$. These findings are a step towards the viable production of fumaric acid through a renewable and environmentally sustainable process. Future work should focus on investigating the use of authentic lignocellulosic hydrolysate.

**Author Contributions:** Conceptualization, R.M.S. and W.N.; methodology, R.M.S.; software, R.M.S.; validation, R.M.S.; formal analysis, R.M.S.; investigation, R.M.S.; resources, W.N.; data curation, R.M.S.; writing—original draft preparation, R.M.S.; writing—review and editing, R.M.S., H.B. and W.N.; visualization, R.M.S.; supervision, W.N. and H.B.; project administration, R.M.S.; funding acquisition, R.M.S. and W.N. All authors have read and agreed to the published version of the manuscript.

**Funding:** This research was funded by the National Research Foundation, grant number MND200609529524.

**Institutional Review Board Statement:** Not Applicable.

**Informed Consent Statement:** Not Applicable.

**Data Availability Statement:** The data presented in this study are openly available at the University of Pretoria Research Data Repository at DOI: 10.25403/UPresearchdata.19883335.

**Conflicts of Interest:** The authors declare no conflict of interest. The funders had no role in the design of the study; in the collection, analyses, or interpretation of data; in the writing of the manuscript, or in the decision to publish the results.

## Abbreviations

The following abbreviations are used in this manuscript:

| | |
|---|---|
| $A$ | Acid concentration (mol L$^{-1}$) |
| ATP | Adenosine triphosphate |
| $C$ | Carbon mole concentration of specie (Cmol L$^{-1}$) |
| CCR | Carbon catabolite repression |
| $e$ | Effluent |
| $F$ | Faraday's constant (96.5 kJ Volt$^{-1}$e-mol$^{-1}$) |
| $f$ | Feed into reactor |
| $j$ | Designation of a specie |
| LH | Lignocellulosic hydrolysate |
| $N$ | Moles of species (Cmol$_i$) |
| NADH | Nicotinamide adenine dinucleotide |
| NADPH | Nicotinamide adenine dinucleotide phosphate |
| $o$ | Inlet |
| $Q$ | Volumetric flow rate (L h$^{-1}$) |
| $r$ | Rate of production (Cmol$_i$ L$^{-1}$ h$^{-1}$) |
| $t$ | Time increment (s) |
| TCA | Tricarboxylic acid |
| $V$ | Liquid volume of the fermenter (L) |
| $V_g$ | Gas volume of the fermenter (L) |

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
