# Peer review of "Rhizopus oryzae for Fumaric Acid Production: Optimising the Use of a Synthetic Lignocellulosic Hydrolysate"

_fermentation, doi:10.3390/fermentation8060278_

Round 1

Reviewer 1 Report

The authors demonstrate improved yields of fumaric acid from Rhizopus oryzae fermentations of soluble sugars (glucose, xylose, and a 50/50 mixture thereof) by careful control of substrate feed rates, in order to decrease metabolic flux to the undesired side product, ethanol. The results were applied to a metabolic flux model which confirmed that optimal fermentation conditions were achieved. Overall, the study was well done, and may provide a basis for an economical platform for fumaric production from lignocellulose hydrolyzates.

            Although the authors achieved a fumaric acid yield of 0.735 g per g of substrate when ethanol production was fully suppressed, the reviewer wonders why the yield was not higher. A  theoretical stoichiometry of Glc + 2 CO2  2 Fumaric acid has not been fully realized in any fermentation, but in the related fungus Rhizopus arrhizus, up to 1.45 mol fumaric acid has been obtained per mol of glucose (i.e., a weight yield of 0.934 g/g  (see Kenealy et al., DOI: 10.1128/aem.52.1.128-133.1986 ). Why has this value not been approached in R. oryzae? Is it due, at least in part, to the production of other, unreported, fermentation products? Could this explain the failure (despite assertions to the contrary, see comment to L172-174 below) to close the fermentation balance?

Specific comments:

L62-67: It might be pointed out here that minor sugars potentially present in lignocellulosic hydrolyzates (such as arabinose and mannose) were excluded from consideration. Can R. oryzae utilize these sugars?

L81-84: Presumably the authors drained the liquid medium from the reactor, leaving the fungal mycelia behind, and then performed the buffer rinse and the addition of fresh medium, correct? This should be explicitly stated.

L87-88: Why was a 50/50 mixture (presumably on a weight basis?) of glucose and xylose used? Most lignocellulose hydrolyzates contain substantially more glucose than xylose, unless the hydrolysis is incomplete.

L99: More detail needed here. Was this tube a commercial product or manufactured in-house? If the latter, where was the polypropylene material obtained? How “rough” was its surface?

L116-117: More detail needed here. How was it determined that the sampling interval achieved this “satisfactory resolution”?

L126-132: It would seem that this chromatographic method would also separate and detect fumaric acid. Why were two separate chromatographic assays conducted? Were they necessary to quantify the minor fermentation products?

L135: Do the authors mean “growth measurements procedure”, rather than “growth procedure”?

L172-174: This description seems to be quantitatively weak. It appears that the authors considered a carbon recovery of 90-110% to be adequate for closing the balance. The reviewer suggests including the actual data for each fermentation run as Supplementary Information. This would allow quantifying the minor fermentation products listed in L123-124, but discussed little thereafter).

L360-361: Actually, from the Figure it appears that fumaric acid production continued for another 24 h, although at a gradually decreasing rate.

L387: Not all, but almost all.

L389-390: This is a reach. What is the basis for this speculation? Weren’t the same calibrated instruments used for the measurements in all runs?

Figure 7 legend: What is meant by compounds that “interact with the medium”? Do the authors mean supplied substrates and extracellular metabolites? This would seem to hold true for all metabolites circled in black, except for glycogen, which is intracellular.

L449: Perhaps an energy imbalance, rather than an energy requirement? An energy requirement suggests a specific energy consuming reaction which, if it exists at all, is not apparent.

L464-465: While the demonstration of improved yields from an LH was established. It would be useful to indicate here what additional improvements would be necessary to move the process toward economic viability -- for example, attaining higher fumaric acid concentration, by using higher substrate concentrations), or extending the work to authentic (rather than synthetic) LHs .

Minor edits:

L23: Change “petrol chemical” to “petrochemical”.

L40: Change “which” to “and”.

L48: Change “since” to “because”. (Also L111, L141, L198, L221, L308).

L74: Change “10 mL” to “Ten mL”.

L100: Silicon or silicone?

L184: Change “parameters” to “variables”.

L239: Change “effecting” to “affecting”.

L379: Change “0.200” to “0.204”. (0.200/0.980 = 0.204).

L410: Replace comma with semicolon.

Reviewer 2 Report

The present manuscript offers a detailed description about the production of fumaric acid from lignocellulosic hydrolisates. The experimental work is well planned, and the results support the previous hypothesis.

However, few grammatical typos need to be corrected before to acceptance

Minor points:

A) Please define “g g-1”

B) Line 62 “The aim of this study” can be replaced by “This study aims to”

C) Line 89 The expression “In order to” is not necessary. Maybe you can start from “To …..”

Additionally the authors also use many times this expression in the manuscript.

D) Line 102 “Prior to” sounds wordy. Maybe you can replace by “before”

E) Line 204. Consider to replace “It therefore offers no benefit” by “Therefore, it does not offer….”

F) Line 358. Consider adding a comma after “In contrast”

G) Authors use to much the word “clearly”

H) Figure 7. Replaced “The metabolic pathways were determine” by “……………..determined” 
